# An olfactory self-test effectively screens for COVID-19

Kobi Snitz[1,16], Danielle Honigstein[1,16], Reut Weissgross [1,16], Aharon Ravia [1,16], Eva Mishor [1,16], Ofer Perl[1,16], Shiri Karagach[1], Abebe Medhanie[1], Nir Harel[2], Sagit Shushan[1,3], Yehudah Roth[3,4], Behzad Iravani [5], Artin Arshamian[5,6], Gernot Ernst [7], Masako Okamoto[8], Cindy Poo[9], Niccolò Bonacchi [9], Zachary Mainen [9], Erminio Monteleone[10], Caterina Dinnella [10], Sara Spinelli[10], Franklin Mariño-Sánchez[11], Camille Ferdenzi[12], Monique Smeets[13], Kazushige Touhara[8], Moustafa Bensafi [12], Thomas Hummel[14], Johan N. Lundström [5,15] & Noam Sobel [1✉]

## Abstract

**Background** Key to curtailing the COVID-19 pandemic are wide-scale screening strategies. An ideal screen is one that would not rely on transporting, distributing, and collecting physical specimens. Given the olfactory impairment associated with COVID-19, we developed a perceptual measure of olfaction that relies on smelling household odorants and rating them online.

**Methods** Each participant was instructed to select 5 household items, and rate their perceived odor pleasantness and intensity using an online visual analogue scale. We used this data to assign an olfactory perceptual fingerprint, a value that reflects the perceived difference between odorants. We tested the performance of this real-time tool in a total of 13,484 participants (462 COVID-19 positive) from 134 countries who provided 178,820 perceptual ratings of 60 different household odorants.

**Results** We observe that olfactory ratings are indicative of COVID-19 status in a country, significantly correlating with national infection rates over time. More importantly, we observe indicative power at the individual level (79% sensitivity and 87% specificity). Critically, this olfactory screen remains effective in participants with COVID-19 but without symptoms, and in participants with symptoms but without COVID-19.

**Conclusions** The current odorant-based olfactory screen adds a component to online symptom-checkers, to potentially provide an added first line of defense that can help fight disease progression at the population level. The data derived from this tool may allow better understanding of the link between COVID-19 and olfaction.

## Plain language summary

From early on in the COVID-19 pandemic, a symptom associated with infection was rapid and often complete loss of the sense of smell. This rendered smell testing a potentially helpful tool in large-scale screening for SARS-CoV-2 infection. We built an online tool (smelltracker.org) that enables assessment of the sense of smell using commonly available household odorants. Initial use by 13,484 participants (462 COVID-19 positive) from 134 countries corroborated that SARS-CoV-2 infection is associated with impaired smell. Moreover, the tool detected infection in the absence of any other symptoms, including subjective loss in smell. Use of this tool may provide an added instrument for screening SARS-CoV-2 infection, and the data generated by the tool may provide for deeper understanding of the brain mechanisms involved with loss of smell associated with COVID-19.

[1] Department of Neurobiology, Weizmann Institute of Science, Rehovot, Israel. [2] Department of Fine Arts, Bezalel Academy of Fine Arts and Design, Jerusalem, Israel. [3] Department of Otolaryngology-Head & Neck Surgery, Edith Wolfson Medical Center, Holon, Israel. [4] Sackler Faculty of Medicine, Tel Aviv University, Tel Aviv, Israel. [5] Department of Clinical Neuroscience, Karolinska Institutet, Stockholm, Sweden. [6] Department of Psychology, Stockholm University, Stockholm, Sweden. [7] Psychological institute, Oslo University, Oslo, Norway. [8] Department of Applied Biological Chemistry, Graduate School of Agricultural and Life Sciences, The University of Tokyo, Tokyo, Japan. [9] Champalimaud Research, Champalimaud Centre for the Unknown, Lisbon, Portugal. [10] Department of Agriculture, Food, Environment and Forestry, University of Florence, Florence, Italy. [11] Rhinology and Skull Base Surgery Unit, Otorhinolaryngology Department, Ramón y Cajal University Hospital, Madrid, Spain. [12] Lyon Neuroscience Research Center, CNRS - INSERM - University Claude Bernard of Lyon, Lyon, France. [13] Faculty of Social and Behavioral Sciences, Utrecht University, Utrecht, the Netherlands. [14] Smell and Taste Clinic, Department of Otorhinolaryngology, TU Dresden, Dresden, Germany. [15] Monell Chemical Senses Center, Philadelphia, PA, USA. [16]These authors contributed equally: Kobi Snitz, Danielle Honigstein, Reut Weissgross, Aharon Ravia, Eva Mishor, Ofer Perl. ✉email: noam.sobel@weizmann.ac.il

The COVID-19 pandemic has wreaked havoc on world order. A necessary tool for effectively dealing with the pandemic is a wide-scale, rapid, and cheap method for screening. In that national medical systems are overloaded as it is, an ideal screening tool would be one that does not entail transportation, dissemination, and processing of a physical test. One alternative that has received attention is the possibility of AI-enabled diagnosis by the sound of coughing, or hoarse voice, that can be recorded on a phone line (AI-Cough)[1]. A second alternative is online subjective self-reported symptom-checkers, that have indeed generated some remarkable results[2,3]. Although both of these approaches satisfy the need for a wide-scale easily-administered screening scheme[4], they both inherently fail at two critical points: One is in individuals who are ill, but not with COVID-19. If an individual has a cough, fever, and a headache, current AI-Cough and symptom-checkers will most likely estimate them to have COVID-19, even when they do not. The second point of failure is individuals who have COVID-19, but experience no apparent symptoms. By definition, such individuals, who constitute ~30% of all those infected[5,6], will go undetected by current AI-Cough and symptom-checkers. Paradoxically, these very same individuals are possibly the most concerning from an epidemiological perspective, as they may unwittingly spread the disease. An alternative to subjective symptom-reporting alone is an objective performance-based test, and the sense of smell provides for a particularly attractive target in this respect. This is for two reasons: First, loss and/or alterations in the sense of smell have been widely recognized as a highly prevalent early symptom of COVID-19[7–10]. Second, most individuals have household odorants readily available to them for testing. Indeed, a large consortium of clinicians and basic scientists known as GCCR (https://gcchemosensr.org) has set out to investigate olfaction in COVID-19, and have convincingly established olfaction as a marker[11–13]. Moreover, in cases where the olfactory loss is subjectively noticed, that alone is a powerful marker of COVID-19[14]. That said, the power of olfaction as a marker in subjectively asymptomatic individuals (namely the most concerning group from an epidemiological standpoint) who have no subjective sense of an olfactory loss, has yet to be tested as far as we know. Because, by definition, subjectively asymptomatic individuals are unaware of any olfactory change or impairment they may have, the test used needs to be one that is particularly sensitive to perceptual alterations that may be subconscious. The recently developed olfactory perceptual fingerprint (OPF) is precisely such an instrument[15,16]. Because it is not a performance measure per se (e.g., a score for detection/discrimination/identification), but rather a perceptual quantifier, it effectively taps into minute subconscious alterations in perception[17,18]. Thus, our hypothesis was that the OPF may allow for accurate classification of individuals who are COVID-19 positive but without symptoms, or COVID-19 negative but with symptoms of (other) disease. Beyond testing this hypothesis, our aim was to generate a convenient online tool that applies this approach. To address these hypotheses we built an online tool for reporting olfactory perception. Here, we report the results from 13,484 participants (462 COVID-19 positive) from 134 countries who provided 178,820 perceptual ratings of 60 different household odorants. We observe that olfactory ratings are indicative of COVID-19 status in a country, significantly correlating with national infection rates over time. More importantly, we observe indicative power at the individual level (79% sensitivity and 87% specificity). Critically, consistent with our hypothesis, this olfactory screen remains effective in participants with COVID-19 but without symptoms, and in participants with symptoms but without COVID-19.

## Methods

**Recruitment**. There was no systematic recruitment. We formed a small international consortium, and each participating lab tried to inform the local media in their country of residence. This resulted is several news stories published in several countries, and these led to dissemination. The success of the publicity varied greatly from country to country due to the resources available to participating labs and to public disposition. Participants were directed to the web-tool at www.smelltracker.org where they consented to participate anonymously in a study that was approved by the Wolfson Hospital Helsinki Committee (Approval #0066-20-WOMC). We note that during the initial reported time period we had 12,800 participants, but of these 780 participants reported only partial data, retaining only 12,020 participants for full analysis. In a reported follow-up we then analyzed an added 1464 participants, culminating at 13,484 participants. This study size reflected a balance between two considerations: Web-based questionnaires on odor intensity and pleasantness gain sufficient power at 198 participants per odorant[16]. We wanted this power for at least 40 odorants, and indeed here report on 42 odorants with more than 198 respondents. In turn, we did not wait for this number of respondents on all possible odorants, as we wanted to report on this in a timely manner, given the progression of the pandemic.

**Web-tool**. We built an online odorant rating tool (www.smelltracker.org). The tool was written in open-source Drupal (drupal.org), and translated into 15 languages by the co-authors who are all respective native speakers. Using the tool, participants first created a unique login to facilitate repeated testing. Otherwise, the tool was completely anonymous to protect user privacy[19]. Next, participants provided details regarding age, sex (female/male), and country of residence (here we made a mistake in that the country pull-down menu did not start from an empty space, but rather from "India". Thus, participants who failed to answer this question were registered as from India by default. For this reason, we are unable to faithfully include India in the country-specific analyses). Next, participants selected five of 71 possible odorants to rate (Supplementary Table 1). We opted for five odorants, rather than a larger number, to strike a balance between increased reliability, where more assessments render more reliable data[20], versus low burden of participation. Each odorant was selected from a separate category with a fixed list of common household odorants (Supplementary Table 1). This list was generated in coordination with the participating labs, each contributing for their native culture in order to assure cultural diversity. Two odorant categories contained odorants with reduced trigeminal components (e.g., vanilla extract), and three categories had increased trigeminal components (e.g., vinegar). Participants made their odorant selections upon first use of the tool, and were then automatically prompted to use the same odorants on subsequent uses. Participants then smelled and rated each odorant using visual-analog scales (VASs) for perceived intensity and pleasantness, namely the primary dimensions of olfactory perception[21]. VASs ranged from very weak to very strong, and from very pleasant to very unpleasant. These scales were coded in the system as ranging from 0 to 100. Participants could smell the odorant as often as they liked, and there was no time limit applied. Following the ratings, participants were asked whether they had been tested for COVID-19 (No; Yes-Pending; Yes-Positive; and Yes-Negative), and whether they are currently experiencing any COVID-19 symptoms (Fever; Cough; Shortness of breath or difficulty breathing; Tiredness; Aches; Runny nose; Sore throat; Loss of the sense of smell; Loss of taste; and No symptoms). We have recently added to the live site questions on vaccinations, but these were not available when this data was collected. Finally, after completing participation, participants were presented with a graph depicting their olfactory perceptual

fingerprint as it related to the average scoring, and if they participated again, the graph depicted the evolution of their perception over time. In addition to the graph, participants were presented with a text informing them whether their perception was within range of most participants, or aberrant. Based on the results now obtained and reported in this manuscript, we have only recently modified the feedback component such that the system now also informs participants to what extent they resemble a person who is COVID-19 positive or COVID-19 negative. Given regulatory restrictions, this is as close as we could get to giving a diagnosis. This extended feedback, however, was not provided to participates reported on in this manuscript.

**Statistics and reproducibility**. All analyses were conducted using Matlab software, and the complete data file allowing full recreation of these results is in Supplementary Data 1. For initial analysis of intensity and pleasantness, we restricted our analysis to 23 odorants that had more than 25 C19+ raters. This gave rise to 46 distributions of ratings, of which only 18 and 33 for intensity and pleasantness, respectively, were normally distributed. Given non-normal distributions, we applied a two-sided Kolmogorov–Smirnov test to all C19+ and C19− intensity and pleasantness comparisons. In the individual odorant follow-up comparisons we estimated effect size using the Eta squared effect size measure[22].

Country-specific correlations between odorant ratings and rates of COVID-19 were calculated as follows: To produce time-series for rates of COVID-19, we conducted two steps: 1. The number of daily cases in each country was obtained from the Johns Hopkins Coronavirus Resource Center[23]. 2. We calculated a 7-day moving average for the dates between March 15, 2020 and September 30, 2020. For national intensity ratings time-series, we conducted three steps: 1. Average intensity ratings of the five odorants were calculated for each entry. 2. Mean intensity ratings were inverted by subtracting them from 100. This was done so that higher values imply greater smell loss. 3. A 5-day moving average was calculated by averaging all ratings in the span of 7 days. We used this moving average to match the cases span. After obtaining these two values, a cross-correlation between the daily ratings and inverse intensity was then calculated (using the xcorr function in Matlab). The cross-correlation analysis resulted in a correlation between the two signals for different lags (between 14-days earlier to 14-days later response) in the inverse intensity signal. The lag that produced the maximal correlation between the two signals was chosen for the analysis. The Pearson correlation between daily cases and lagged inverse intensity was calculated. Daily cases time-series, inverse intensity signal and lagged inverse intensity signal are shown the related figure.

Receiver operating curves (ROCs) were calculated using standard technique[24]. We used a moving cutoff point on a continuous scale, and at each point measured the true positive (TPR) and false positive (FPR) ratios which result from selecting that cutoff. All confidence intervals in ROC plots were calculated using a 1000 iteration bootstrapping of the scores. To compare between ROCs, we used a non-parametric test based on the AUC of the curves[25].

**Olfactory perceptual fingerprints**. Individual olfactory perception is typically characterized using performance-based measures, such as olfactory detection, discrimination and identification[26,27]. An alternative is not to characterize performance, but rather characterize how the world smells to an individual. Such characterizations have been termed olfactory perceptual fingerprints (OPFs), and their typical derivation relies on the perceptual distance matrix for a set of odorants[15,16]. One version of the OPF is the descriptor-based OPF. Here, an individual is characterized by how he/she applies a set of descriptors to a set of odorants. Given M odorants and N descriptors, for each participant m, for each odorant we calculate the difference between their rating along a descriptor, versus the group mean for that same odorant and same descriptor. Thus, each participant is initially described as a matrix $P_m$ where each entry $p_{i,j}$ is the difference between their perceptual rating of an odorant i along a descriptor j and the group mean of the same odorant and descriptor.

This yields M relative scores along each of N descriptors. We then average M relative scores for each of the N descriptors. This in a N dimensional representation of the individual $\widetilde{P_m}$, where each entry in $\widetilde{P_m}$ is by Eq. (1):

$$(\widetilde{p_m})_j = \frac{\sum_{i=1}^{M} p_{i,j}}{M} \tag{1}$$

In the current study this is simplified, as we have five odorants (self-selected) and two descriptors, namely intensity and pleasantness. This yields five relative intensity and five relative pleasantness scores. We then average the five intensity differences and five pleasantness differences for each participant, retaining two numbers that represent that participant in a two-dimensional space. The advantage of this descriptor-based approach is that it allows us to directly compare individuals, who selected different odorants. The only perquisite for generating the calculation is that a sufficient number of individuals (although not necessarily the individuals under comparison) rated a given odorant-descriptor pair, so that we have a valid mean entry for that pair. This combination of conditions renders this method ideal for the current data. We acknowledge that this measure may be weakened by cultural/geographical variability in olfactory perception[28]. We note however, that pleasantness reflects the primary physical dimension in odorant structure[21], and it is the primary dimension of olfactory perception[29]. Therefore, cross-cultural variability in odorant pleasantness is far lower than commonly thought[21,30,31], and typically overestimated because of a few canonical outlying odorants. Finally on this front, we will stress that to the extent that this is a shortcoming, it is one that can only weaken our result, not strengthen it or generate an artifactual outcome.

**Reporting summary**. Further information on research design is available in the Nature Research Reporting Summary linked to this article.

## Results
**Olfactory perception indicates on levels of COVID-19 infection at the population level**. Between the dates of March 25th 2020 and September 23rd 2020, we collected data from 12,020 individuals (7189 Women, mean age = 44.32 ± 14.28, 4831 Men, mean age = 45.23 ± 15.29) (Fig. 1a), residing in 134 countries (Fig. 1b, c). Of these, 348 participants reported positive COVID-19 test results (C19+), 400 participants reported negative test results (C19−), and the COVID-19 status of the remaining 11,272 participants was unknown (C19-UD) (Fig. 1d). The presence of at least one disease symptom was reported by 91.1% of C19+ participants, 55.3% of C19− participants, and 55% of C19-UD participants (Fig. 1e). Participants could use the online tool repeatedly, yet 10,103 participants (84%) used it only once, 1130 participants (9.4%) used it twice, and the remaining 6.6% of participants used it various number of times (Fig. 1f). In total, the 12,020 participants provided 171,500 ratings applied to 60 different odorants (i.e., 11 odorants were never rated) (the entire raw data file is available in Supplementary Data 1).

To probe for any gross differences in olfactory perception between C19+, C19−, and C19-UD, we plotted their overall odorant intensity and pleasantness estimations (Fig. 2a–c).

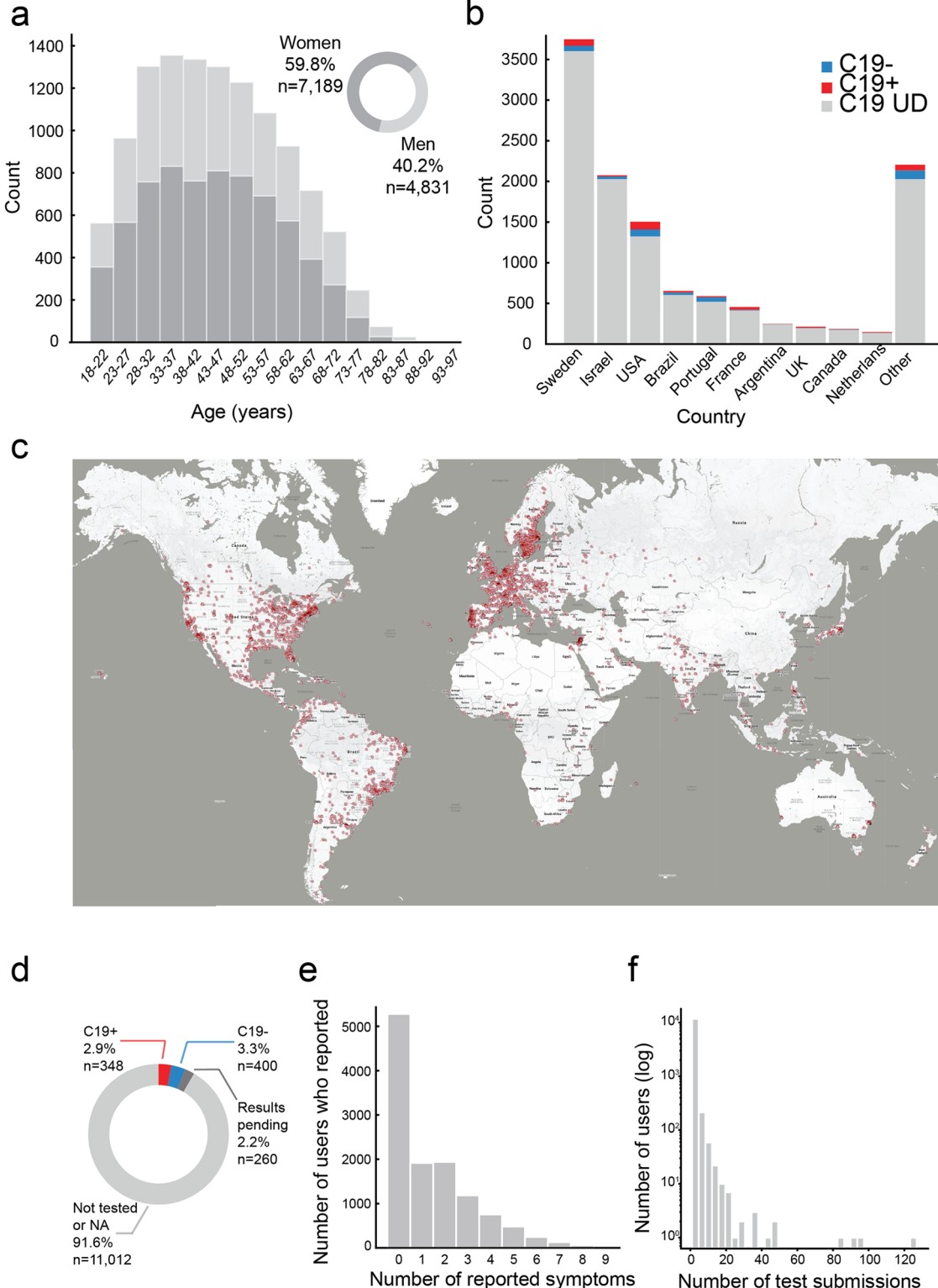

**Fig. 1 Characterization of 12,020 participants. a** Histogram of age and gender distribution of participants. **b** Histogram of number of participants and their COVID-19 status from the ten highest-participation countries (see comment on India in "Methods" section). **c** Map of geographical distribution of respondents, each dot is a participant, overlapping dots not shown to maintain clarity. **d** Pie chart of the distribution of C19+, C19−, and C19-UD in the sample. **e** Histogram of the distribution of number of somatic symptoms reported by participants. **f** Histogram of the distribution of number of submissions per participant.

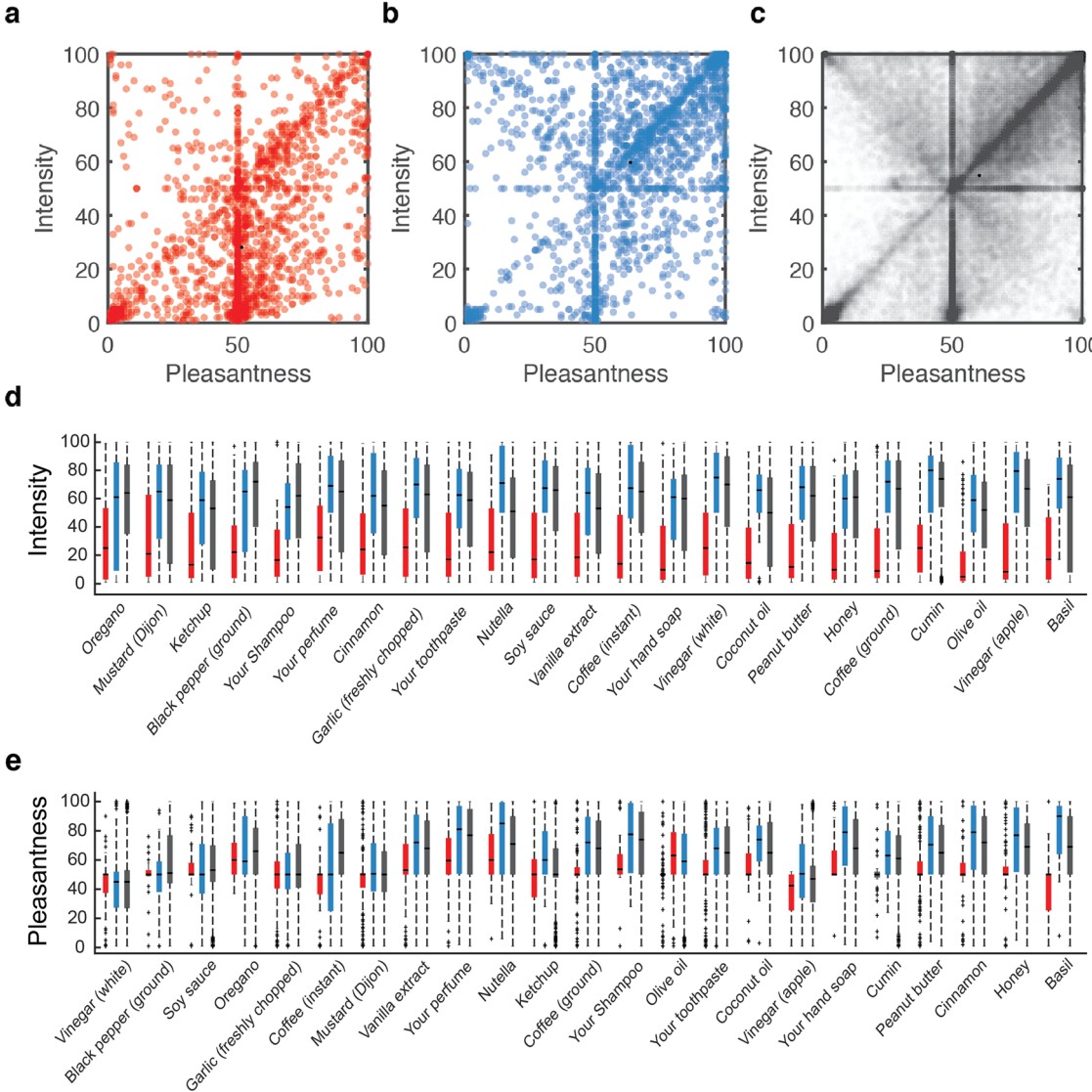

**Fig. 2 Olfactory perception indicates on levels of COVID-19 infection at the population level. a–c** Each dot is an odorant rating, aligned for its pleasantness and intensity estimates: **a** All ratings from C19+ participants ($n = 2670$ ratings). **b** All ratings from C19− participants ($n = 2580$ ratings). **c** All ratings from C19-UD participants ($n = 80,500$ ratings). **d** Intensity estimates for the 23 odorants that were each rated by at least 25 C19+ participants, ordered by effect-size from low (left) to high (right). **e** Pleasantness estimates for the 23 odorants that were each rated by at least 25 C19+ participants, ordered by effect-size from low (left) to high (right). C19+ in red, C19− in blue, and C19-UD in black. **d, e** The central mark in the box indicates the median. Bottom and top edges of the box indicate the 25th and 75th percentiles, respectively. The whiskers extend to the most extreme data points not considered outliers. Outliers are plotted individually using the "+" symbol.

Consistent with previous reports[2,9,32–34], these gross plots revealed pronounced differences between groups in intensity (two-sided Kolmogorov–Smirnov, $D = 0.45$, $p = 3.96e−233$, corrected) and pleasantness ($D = 0.31$, $p = 2.36e−114$, corrected) (Fig. 2a–c). To provide a finer-grain view of this, we examined individual odorants, restricting our analysis to odorants that were rated by at least 25 C19+, thus retaining 23 odorants. Because a Kolmogorov–Smirnov normality of distribution test revealed that data for some of the odorants was not normally distributed, we proceeded with a non-parametric approach. A Chi squared test comparing C19+ and C19− ratings was significant for each of the 23 odorants (all Chi Square > 7.6, all $p < 0.0058$, all Eta squared effect size > 0.08) (Fig. 2d), and the same test on pleasantness ratings was significant for 17 of the 23 odorants (all Chi Square > 8.69, all $p < 0.0188$, all Eta squared effect size > 0.02)

(Fig. 2e) (we note that replicating this analysis using a parametric analysis of variance yielded nearly identical results).

Having observed that, consistent with previous reports, these gross measures of perception implied altered olfaction in COVID-19, we asked whether they were related to the COVID-19 status over time in the different countries where we collected data. Consistent with an initial analysis[35], we concentrated on odorant intensity estimates in this global analysis, and limited this to countries with at least 250 respondents of which at least ten were formally diagnosed. This limited us to eight countries, where country-specific rates of COVID-19 infection over time were obtained from The Johns Hopkins University Coronavirus Resource Center[23]. We observed a significant relationship between overall group-level odorant intensity ratings and daily rates of COVID-19. More

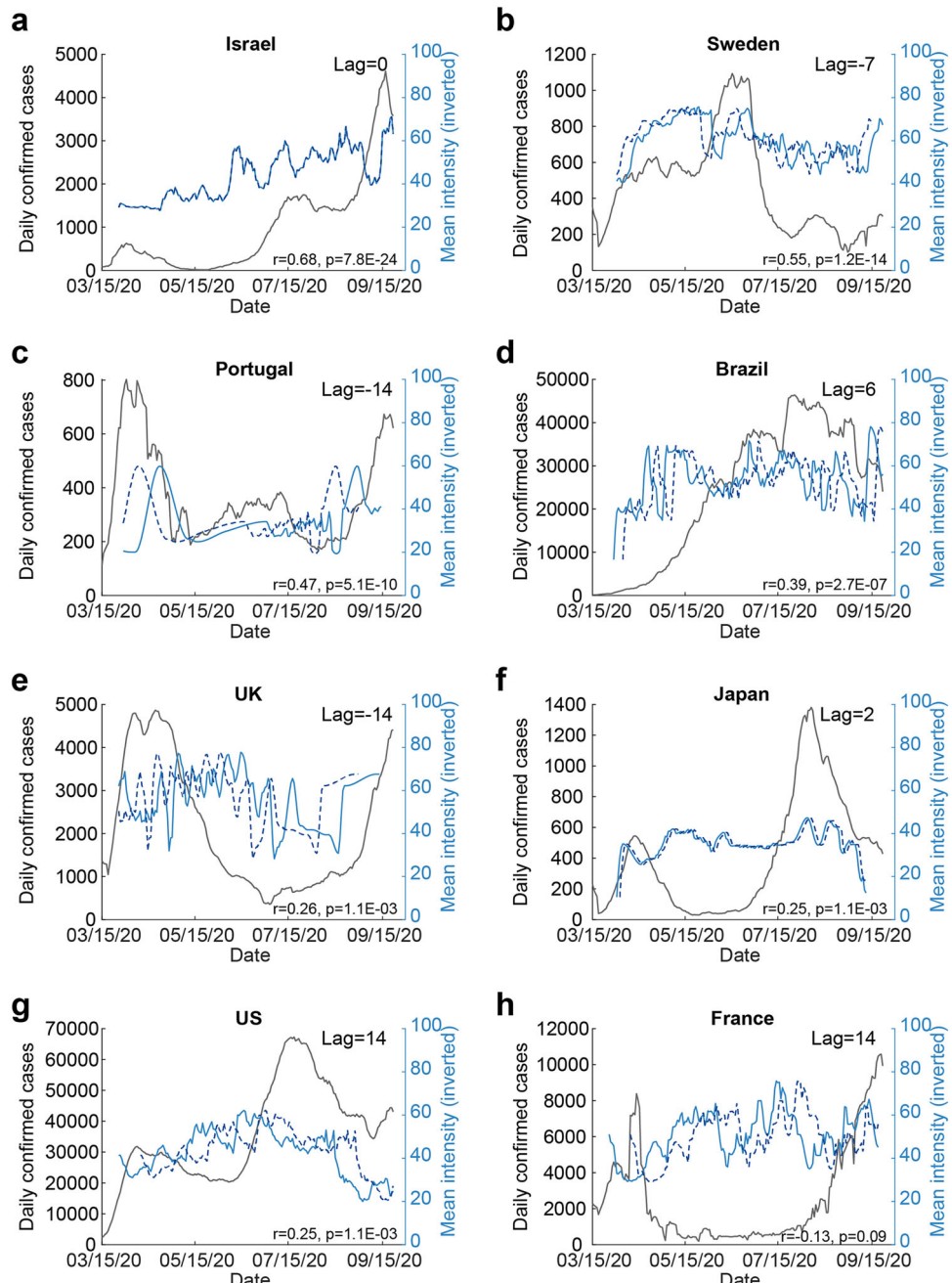

**Fig. 3 Odorant intensity estimates correlated with national COVID-19 infection rates over time. a** The correlation between intensity estimates and national levels of COVID-in Israel. Blue line: mean daily additive inverse intensity ratings. Dashed blue line: shifted additive inverse intensity time-series, after finding the peak lag using cross correlation (see "Methods" section). Black line: number of daily confirmed cases in each country. Note that when the lag is close to zero, then the dashed and solid blue lines align and overlap. All other panels the same as **a** but for: **b** Sweden. **c** Portugal. **d** Brazil. **e** United Kingdom. **f** Japan. **g** United States. **h** France.

specifically, mean intensity ratings and COVID-19 prevalence were significantly correlated in seven out of eight countries (Sorted by Pearson correlation FDR corrected: Israel: $n = 2734$, $r = 0.68$, $p < 0.0001$; Sweden: $n = 6133$, $r = 0.55$, $p < 0.0001$; Portugal: $n = 685$, $r = 0.47$, $p < 0.0001$; Brazil: $n = 764$, $r = 0.39$, $p < 0.0001$; UK: $n = 290$, $r = 0.26$, $p = 0.0011$; Japan $n = 290$, $r = 0.25$, $p = 0.0011$; USA: $n = 2276$, $r = 0.25$, $p = 0.0011$; France: $n = 655$, $r = −0.14$, $p = 0.09$) (Fig. 3). Although the country-specific sample sizes are not overwhelming in this sub-analysis, these results imply that olfactory testing can augment symptom-tracking[36] to aid country-level rapid policy decisions related to the spread of COVID-19.

**Olfactory perception indicates on COVID-19 at the individual level**. Having observed that olfaction provides for an indication on levels of COVID-19 infection at the country level, we next asked whether it can provide an indication on COVID-19 in an individual. Given the primacy of intensity estimates at the populational level, we initially concentrated on those. We continued with the 23 odorants that were rated by at least 25 C19+ participants. These ranged in usage from the odorant "black pepper ground" that had the smallest number of positively diagnosed raters at only 26 C19+ and 43 C19− participants, to the odorant "your toothpaste" that was rated by as much as 336 C19+ and 330 C19− participants. We then plotted receiver

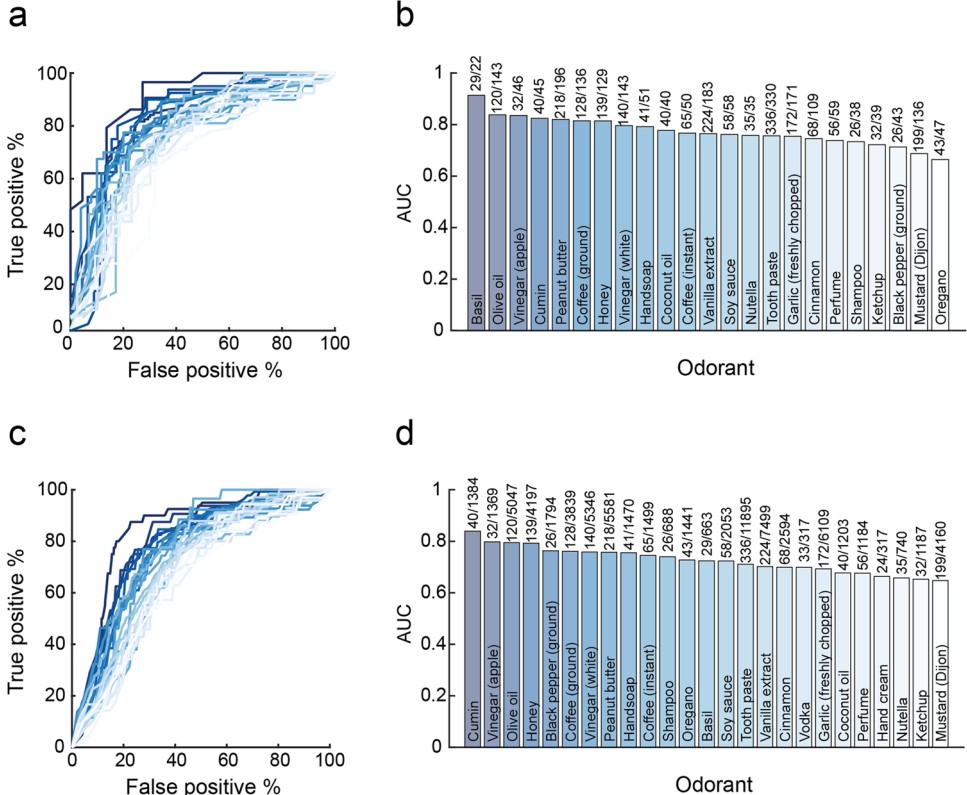

**Fig. 4 Single odorant intensity estimates indicate on COVID-19 at the individual level. a** Receiver-operator curves (ROCs) based on intensity estimates of 23 odorants obtained from C19+ vs. C19− participants. **b** The area under the curve (AUC) for each odorant, with the number of C19+/C19− participants above each bar. We note that these bars reflect a single value, the AUC of all available data, which has no error associated with it. **c** ROCs based on intensity estimates of 23 odorants obtained from C19+ vs. C19− and C19-UD combined participants. **d** The AUC for each odorant, with the number of C19+/C19− and C19-UD participants above each bar.

operator curves (ROCs) to gauge the potential classification power of intensity estimates associated with each of these odorants. We observed meaningful classification potential, with seven odorants generating ROCs with area under the curve (AUC) greater than 0.8 (Fig. 4a, b). Most remarkable was the odorant Basil, with an AUC of 0.91. ROC classification success can be estimated at different true positive (sensitivity) rates selected by the observer. For context on sensitivity, we consider antigen tests, as these are promoted as a cheaper and more available alternative to RT-PCR, and have been directly compared to the latter. Antigen results vary widely[37], and their results are environmentally dependent[38]. At the lower end, antigen tests have obtained sensitivity of 30.2% in direct comparison to RT-PCR[39]. At the upper end, antigen tests have obtained specificity of 100% and sensitivity of 79.6% in direct comparison to RT-PCR[20]. In the case of the above Basil-derived ROC, at a true positive rate of 62% (namely on par with lower-bound estimates for RT-PCR itself[40]), we retain a remarkably low 5% false positive rate, translating to 62% sensitivity and 95% specificity (95% confidence on sensitivity: 43–80%, 95% confidence on specificity: 75–100%, $p = 0.03$ corrected, PPV = 0.94, NPV = 0.65, Matthews Correlation Coefficient = 0.58) in detecting COVID-19 using intensity estimates of the odorant Basil alone. In turn, at a more conservative true positive rate of 79%, we retain a modest 13% false positive rate, translating to 79% sensitivity and 87% specificity (95% confidence on sensitivity: 63–92%, 95% confidence on specificity: 67–97%, $p = 0.0043$ corrected, PPV = 0.88, NPV = 0.76, Matthews correlation coefficient = 0.65). In other words, using this approach we correctly classify 42 of 51 COVID-tested individuals who smelled Basil. One may note that selecting a 79% true

positive rate still implies a 21% false negative rate, and this is potentially costly. This balance reflects a question of policy, and favoring a low false negative rate may be preferred in this pandemic[41]. With that in mind, we observe that if we select a 97% true positive rate, we incur a 27% false positive rate, reflecting 97% sensitivity and 73% specificity (95% confidence on sensitivity: 79–100%, 95% confidence on specificity: 50–89%, $p < 0.00001$ corrected, PPV = 0.63, NPV = 0.85, Matthews correlation coefficient = 0.342). This reflects a 3% false negative rate, well within the optimal goal of testing[41].

A limitation of the above result is that it relies on a restricted subset of our data, namely tested individuals who also smelled Basil. To overcome this, we recalculated ROCs, now comparing between C19+ and all other participants (C19− combined with C19-UD). In other words, we assumed that untested individuals are not sick with COVID-19. Although this comparison may be weakened by unidentified C19+ individuals within the C19-UD cohort (i.e., this works against us), it allows for greater sample sizes per odorant. Once again, we observe meaningful ROCs, with four odorants generating ROCs with AUCs greater than 0.79 (Fig. 4c, d). The strongest, namely Cumin, had an AUC of 0.83. This implies that we could use intensity estimates of the odorant Cumin alone to identify COVID-19, and at a true positive rate of 77%, we retain a 16% false positive rate, translating to 77% sensitivity and 84% specificity (95% confidence on sensitivity: 62–88%, 95% confidence on specificity: 50–89%, $p < 0.00001$ corrected, PPV = 0.11, NPV = 0.99, Matthews correlation coefficient = 0.256). Although Cumin had the largest AUC in this analysis, it was rated by 1424 participants overall, and of these only 40 participants were C19+. Similar numbers were evident in

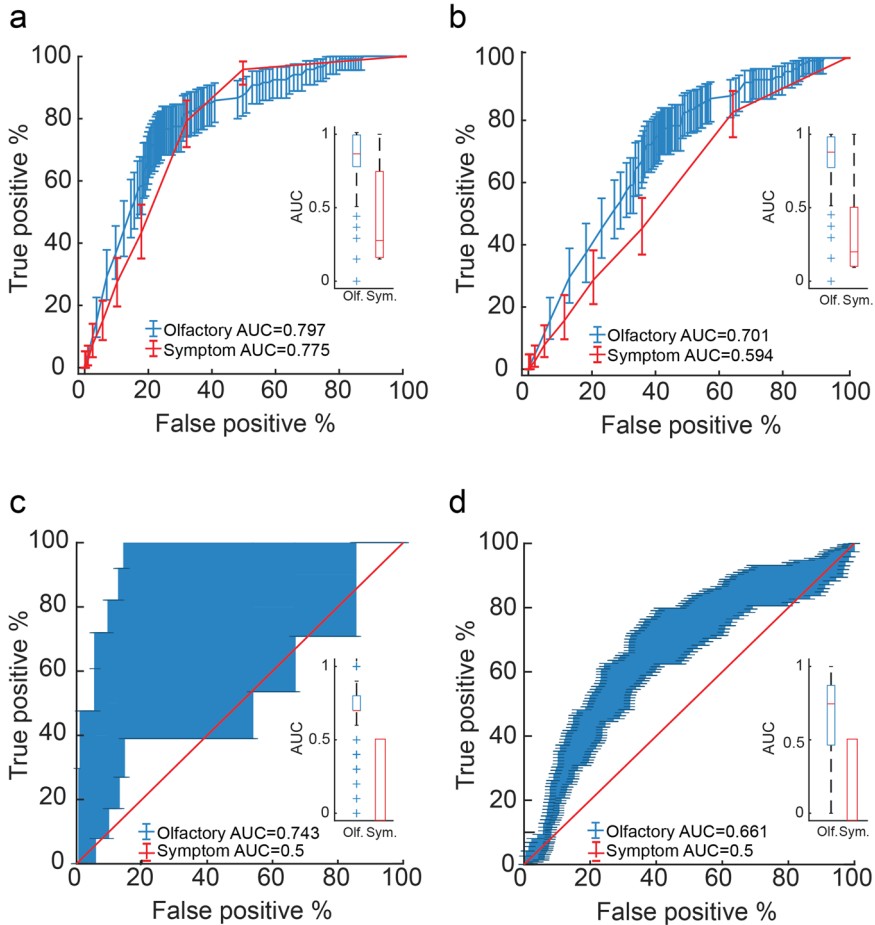

**Fig. 5 Olfactory testing is more effective than symptom checking. a** Receiver-operator curves (ROCs) for all participants who smelled Olive Oil ($n = 5167$ participants), based on odor intensity (blue) or reported symptoms (red). For all panels: the inlay reflects mean ROC area under the curve (AUC). The central mark in the box indicates the median. Bottom and top edges of the box indicate the 25th and 75th percentiles, respectively. The whiskers extend to the most extreme data points not considered outliers. Outliers are plotted individually using the "+" symbol. **b** ROCs for all participants who smelled Olive Oil and had symptoms ($n = 2627$ participants), based on odor intensity (blue) or reported symptoms (red). **c** ROCs for all asymptomatic participants ($n = 7740$ participants), based on a classifier using OPFs (blue) or reported symptoms (red). We note no variance in the symptom-based box (red) as all participants were asymptomatic. **d** Same as in **c**, but for a later-collected independent set of 1464 participants, with a significantly higher proportion of C19+. Note that the reported symptoms have zero variance in **c** and **d** because these are all completely asymptomatic participants. The difference in variability between **c** and **d** is because **c** has 33 asymptomatic C19+ participants, yet **d** has 114 asymptomatic C19+ participants.

the second-best AUC, namely Apple Vinegar. However, the third best AUC, namely Olive Oil, was rated by 5167 participants, of which 120 were C19+. This large number of raters merits concentrating on Olive Oil as a model odorant for what this single-odorant approach might achieve. We observe that Olive Oil had an AUC of 0.79, and if we use intensity estimates of Olive Oil alone to identify COVID-19, at a true positive rate of 77%, we retain a 28% false positive rate, translating to 77% sensitivity and 72% specificity (95% confidence on sensitivity: 68–84%, 95% confidence on specificity: 70–73%, $p < 0.00001$ corrected, PPV = 0.06, NPV = 0.99, Matthews correlation coefficient = 0.16).

These results raise the tantalizing possibility of rapidly detecting COVID-19 by rating the perceived intensity of one odorant, such as Basil or Olive Oil, alone. Moreover, we observe that if we generate ROCs for the same 5167 participants that smelled Olive Oil, but base them on their subjective reported symptoms (fever, cough, etc., including subjective loss in sense of smell and taste)[2] rather than on their objective sense of smell, we obtain a ROC AUC of 0.77 (Fig. 5a). Using this symptom-based ROC AUC of 0.77, at a true positive rate of 79%, we retain a 32% false positive rate, translating to 79% sensitivity and 68% specificity (95% confidence on sensitivity: 71–85%, 95% confidence on specificity: 67–70%,

$p < 0.00001$ corrected, PPV = 0.05, NPV = 0.99, Matthews correlation coefficient = 0.15). Although a lower value, this result is not significantly different from the olfaction-based ROC AUC of 0.79 obtained in the same individuals (StAR analysis for comparing ROCs[25], AUC difference = 0.02, $p = 0.29$).

Does this minimal difference (Fig. 5a) imply that single-odorant olfactory testing has no advantage over symptom checking? Although some symptom checkers have reported even stronger results than those we obtain here[3], there are two critical points where symptom-checking alone all-out fails. One such point is with individuals who all have somatic symptoms such as fever, etc., but do not have COVID-19. Here symptom-checkers cannot avoid false positives. To address this specific point, we restricted our analysis to only participants that reported symptoms. This retained 115 C19+ symptomatic participants and 2512 other, yet also symptomatic participants who smelled Olive Oil. Here symptoms alone gave rise to a ROC AUC of 0.59. In turn, we observe that the Olive Oil derived ROC in these same participants has an AUC of 0.70, which is significantly better (StAR analysis for comparing ROCs[25], AUC difference = 0.11, $p = 0.00022$) (Fig. 5b). Thus, if we use intensity estimates of Olive Oil alone to identify COVID-19 in uniformly symptomatic populations, at a true positive rate of 75%, we retain a

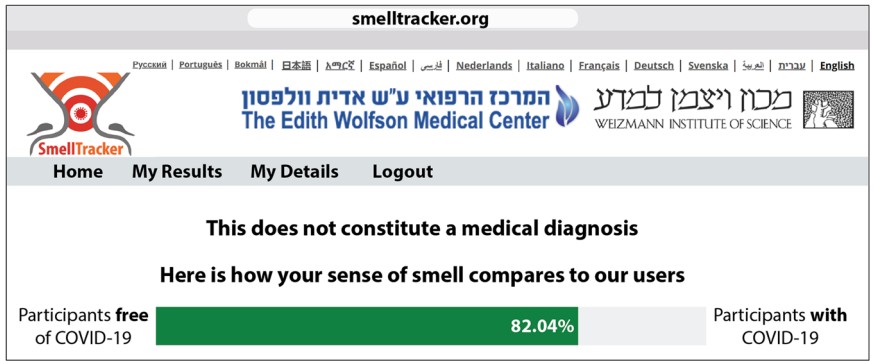

**Fig. 6 Implementation through informative feedback.** At the end of a ~5 min interaction, participants are informed as to what extent their sense of smell resembles a C19+ or C19− participant. The above depiction is an anecdotal actual case of an individual who was C19+, but completely subjectively unaware of any olfactory loss or impairment. Nevertheless, based on the olfactory perceptual fingerprint (OPF), our algorithm determined that this individual was 82.04% C19+. This implies a useful implementation of our tool.

40% false positive rate, translating to 75% sensitivity and 60% specificity (95% confidence on sensitivity: 66–82%, 95% confidence on specificity: 58–62%, $p < 0.00001$ corrected, PPV = 0.08, NPV = 0.98, Matthews correlation coefficient = 0.14) (Fig. 5b).

**Olfactory perceptual fingerprints perform independently of symptom checkers.** A second point where symptom-checkers fail is with individuals who have COVID-19, but absolutely no somatic symptoms (including no sense of altered olfaction). Here symptom-checkers cannot avoid false negatives. In addressing this specific point using the single-odorant approach, we are currently restricted by power. This is because we have only one participant who was C19+, smelled olive oil, and was completely asymptomatic (this reality may change for the better if usage of this tool increases). To overcome this, we use a method that allows us to compare subjects across odorants, thus retaining all 33 C19+ completely asymptomatic and 7707 other completely asymptomatic participants in our study. The method, termed the descriptor-based olfactory perceptual fingerprint (OPF) is described in the "Methods" section. This method does not necessitate that all participants under comparison smell the same odorants, but rather only that each odorant smelled was also smelled by a sufficiently large cohort to provide for a stable mean rating. The method then calculates distances from the mean, and uses these distances to characterize the rater. To now further gauge whether the OPFs are merely detecting symptoms or specifically detecting COVID-19, we restrict our analysis to asymptomatic participants only. We then applied an SVM classifier to the OPFs, training the classifier on one set of 46 participants (23 C19+ and 23 C19−), and then testing on a different set of 260 participants (10 C19+ and 250 C19−). The size of the sets was selected from the available data to create a balanced training set and non-skewed testing set. We repeated this process 500 times for different selections of training and testing sets (we assured that the testing set was made of raters who participated only once, so as to prevent any double-dipping). The symptoms-based ROC AUC in these participants was 0.5 (by definition, as they had no symptoms). In turn, the OPF-derived ROC AUC was 0.74. Using this symptom-based ROC AUC of 0.74, at a true positive rate of 70%, we retain a 31% false positive rate, translating to 70% sensitivity and 69% specificity in a cohort of all completely asymptomatic individuals (95% confidence on sensitivity: 39–99%, 95% confidence on specificity: 26–37%, $p < 0.001$, PPV = 0.08, NPV = 0.98, Matthews correlation coefficient = 2.65e−5) (Fig. 5c). This value is unsurprisingly higher than the symptom-based ROC AUC of 0.5 obtained in the same individuals, but this difference is not statistically significant because it reflects a difference of 10 C19+ individuals only (StAR analysis for comparing ROCs[25], AUC difference = 0.23, $p = 0.15$) (Fig. 5c). We therefore next address this limitation in power.

The above results imply a potentially effective screen for COVID-19, valid in participants with symptoms but without COVID-19, and in participants with COVID-19 but without symptoms. However, this framework has two limitations: First, because this set of results included modeling and testing using the same data set (although obviously not the same participants), it would gain conceptual power from testing on an independent set of data. Second, whereas we could retain many participants for the test of symptomatic C19+ vs. symptomatic C19− participants, we had much fewer participants for the asymptomatic C19+ vs. asymptomatic C19− participants. This is because we had very few completely asymptomatic C19+ participants (our strict criteria were to exclude for even just one somatic symptom reported). To address both of these limitations, we set out to download an additional ensuing data-set from our web-tool. In addition to testing the exact same algorithm in an independent data-set, the advantage of this is that this later (calendar-wise) data-set corresponds to an increase in testing frequency world-wide, thus making for a much higher proportion of asymptomatic C19+ participants. We downloaded data from asymptomatic users only, who participated between 03/25/2020 and 04/10/2021. This provided for an added 114 C19+ and 1350 other completely asymptomatic individuals, who provided an added 7,320 olfactory ratings (Supplementary Data 2). We calculated OPFs exactly as before and applied the same classifier trained on the original previous data set. We observed a ROC AUC of 0.66, where at a true positive rate of 67%, we retain a 36% false positive rate, translating to 67% sensitivity and 64% specificity in an all-asymptomatic cohort (95% confidence on sensitivity: 58–76%, 95% confidence on specificity: 34–39%, $p < 0.001$, PPV = 0.13, NPV = 0.96, Matthews correlation coefficient = 6.4e−7) (Fig. 5d). This result is indeed significantly better than the symptom-based ROC AUC of 0.5 obtained in the same individuals (StAR analysis for comparing ROCs[25], AUC difference = 0.16 $p < 0.001$) (Fig. 5d). In other words, a larger and independent data set confirmed the effectiveness of our model. With these results in hand, we modified the feedback component of the online tool, and this now allows for a more effective implementation. More specifically, a user can now participate using any 5 of the 71 available odorants, and after completing the perceptual estimates, we inform the user to what extent (in percentages) he/she resembles a C19+ or a C19− individual (Fig. 6). A person who has high resemblance to C19+ individuals, should likely minimize contact with others, and seek a formal test.

## Discussion
We found that an olfactory self-test using household odorants performs well at detecting COVID-19, and that it continues to perform more modestly but still significantly in participants with

symptoms but without COVID-19, or most critically, with COVID-19 but without symptoms. In this respect, this result provides a valuable addition to symptom-checkers.

That olfaction serves as such a strong indicator of COVID-19 suggests that the olfactory impairment may be related to some fundamental aspect of this disease. Nevertheless, the mechanism by which the SARS-CoV-2 virus or the COVID-19 disease impact olfaction remains unknown[42]. The effect may be peripheral, reflecting epithelial inflamation[43], or central, reflecting impact on the olfactory brain. Evidence for the latter can be seen in cases of COVID-19 associated olfactory pathway neuropathey[44], and COVID-19 associated olfactory bulb edema[45] and atrophy[46,47]. If the virus reaches the brain through olfactory pathways[48], this is likely not via olfactory receptor neurons (ORNs), but rather through sustentacular non-neuronal supporting cells[49,50]. Like ORNs, these provide for a direct path from the intranasal periphery into the brain, and may underlie neurological aspects of the disease[51]. Moreover, given the intimate link between olfaction and respiration in the brain[52], the olfactory path may enable the virus access to respiratory centers, thus making for a neural component in the respiratory failure associated with the disease[53]. The current study does not provide for any mechanistic insight in this respect, although the apparent difference between specific odorants in their usefulness for classification (e.g., the unique power of Basil) may reflect an avenue worthy of investigation in this respect.

To readers not versed in human olfaction, it may be unclear how our algorithm retains performance across countries, regions, and cultures. The power of the olfactory perceptual fingerprint is that it relies on the relative differences from the overall mean rating. The mean is not regional, it's over all the data. Thus, if there was some huge cultural variability in these two ratings, that would weaken our measure. However, in contrast to some popular notions, these two particular ratings (intensity and pleasantness) are in fact quite stable across large populations, and across very diverse cultures[15,16,21,30,31]. Although some odors have gained notoriety as outliers in this respect (guava, cilantro, and durian), they have gained notoriety for exactly that reason: they are outliers. Otherwise we see very high agreement on these two ratings[15,16,21,30,31].

This study has several limitations. First, we should clearly acknowledge that this study is firstly a basic science effort, and not a clinical effort in the classic sense. For example, we do not address WHO guidelines or Target Product Profile[54], nor do we address any Gold-Standards of diagnostics. What we do here is to find that the OPF allows classification even in asymptomatic individuals, and we provide an online tool that may assist in this. Much more work is needed for this effort to satisfy clinical standards. Second, although our overall data set is large, several of our analyses relied on restricted subsets that reduce power. Third, participants were self-selected, and this may have introduced bias. That said, we fail to identify a selection bias pattern that might underlie our effects. For example, we observe that only 4031 participants (33.5%) reported a subjective loss of smell, so it was not the case that just individuals who felt they lost their sense of smell used this tool. Similarly, we observe that participants were not evenly distributed across countries, nor across sexes: more women participated than men. Although some studies have suggested that women may have a better sense of smell than men[55], we fail to see how this could influence the OFP-based analysis. This, again, is because the OFP is a relative measure, and not an absolute measure of performance per se. Therefore, these sampling biases, although unwanted, likely did not influence the reported result. Fourth, we have no formal verification for the COVID-19 testing reported by our participants. Here too, however, any misrepresentations could have only

weakened our results, as they would have only introduced added noise. Relatedly, we note that even if we had formal verification of RT-PCR tests of our participants, we nevertheless retain an upper bound on measured performance, as RT-PCR itself is not perfect. In other words, we observe that what we are predicting in this study is RT-PCR results, and not SARS-CoV-2 infection itself. Thus, again, our true performance level may be lower or higher than we appreciate. Finally on this front, in those diagnosed positive, we do not have a time point for the diagnosis. Given that the clinical sensitivity of RT-PCR decreases with days post symptom onset, all the way down to 30% at Day 21[56], this information is important towards characterizing the value of olfactory testing. In this respect, we also observe that given the long-term persistence of olfactory impairments in COVID-19[57], our tool can be effective at detecting initial infection, but it is inappropriate for gauging continued infection risk from C19+ individuals.

Beyond all of the above, the COVID-19 pandemic is rapidly evolving, in fact far faster than the process of publishing a manuscript. For example, when we collected the first portion of the data reported here, COVID-19 tests were barely available, yet when we collected the second portion of the data, tests were very common, which made for the higher proportion of COVID-19+ participants. In turn, even at the second collection stage, vaccines were not yet widely available, yet now they are. Anecdotal evidence suggests reduced olfactory loss in vaccinated individuals that nevertheless contract the disease. If these anecdotes materialize, this will render our tool not useful for the vaccinated. This, however, awaits formal results on olfaction in sick yet vaccinated individuals. Despite all these limitations, using a single odorant and a simple measure, namely intensity estimates, provided for a remarkably powerful tool. Although this potential speed of testing (less than 30 s) and simplicity of analysis are both attractive features, the applicability of this approach will be restricted to limited settings. This is primarily because of the susceptibility of this approach to interference. If a cognizant adult self-tests using a single odorant, they will easily understand the test, and may then influence it, whether knowingly or unknowingly. This limitation is overcome by the OPF. Naïve users have no intuition for this measure and how it is calculated. Moreover, the OPF is particularly sensitive to shifts in olfactory perception that do not entail a universal reduction in intensity perception alone, and such shifts may indeed be prevalent in COVID-19[58]. Finally, the OPF was significantly more effective than symptom checkers in the entire cohort. Therefore, despite this test taking longer than single-odorant rating (about 3 min for four odorants), it may be more useful. Notably, these approaches are not mutually exclusive. The on-line interaction can remain the same, and the analysis for a one-time user can favor odorant intensity, yet the display and analysis of repeated users can rely on OPFs. Such repeated tests may gain added power[59], and provide the basis of a testing-regimen, a critical aspect of population-level curtailment[60]. We think this is a rare case where something so utterly simple may nevertheless prove to be valuable.

## Data availability
All raw data are available for download in Supplementary Data 1 and 2. These files allow complete reanalysis except the geographical mapping in Fig. 1, as location data was stripped to protect privacy. Moreover, each figure is associated with a source-data file, entitled Supplementary Data 3–7, allowing recreation of the figure.

## Code availability
All code used in this manuscript is available for download at https://gitlab.com/snitz/smelltracker_article, and in a public repository[61]

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

## Acknowledgements

This study was supported by a Weizmann CoronaVirus Emergency Fund grant from Miel de Botton and a Weizmann CoronaVirus Emergency Fund grant from Sonia T. Marschak. The funders had no role in the design, analysis, or reporting of this manuscript.

## Author contributions

Developed concepts: K.S., D.H., R.W., A.R., E.M., O.P., S.K., N.H., S.S., Y.R., B.I., A.A., G.E., M.O., C.P., N.B., Z.M., E.M., C.D., S.S., F.M.S., C.F., M.S., K.T., M.B., T.H., J.L., N.S., Built web-tool: K.S., D.H., N.H., R.W. Translated to native language: R.W., E.M., S.K., A.M., B.I., A.A., G.E., M.O., C.P., N.B., Z.M., E.R.M., C.D., S.A.S., F.M.S., C.F., M.S., K.T., M.B., T.H., and J.N.L. Analyzed data: K.S., R.W., and A.R.; E.M. Figures and visualization: K.S., R.W., A.R., E.M.; O.P. Wrote first draft of paper: K.S., R.W., A.R., E.M., N.S. Edited final draft: K.S., D.H., R.W., A.R., E.M., O.P., S.K., N.H., S.S., Y.R., B.I., A.A., G.E., M.O., C.P., N.B., Z.M., E.M., C.D., S.S., F.M.S., C.F., M.S., K.T., M.B., T.H., J.L., and N.S.

## Competing interests

The authors declare no competing interests.
