## [Peer Review File · Communications Medicine]

Reviewers' comments:

Reviewer #1 (Remarks to the Author):

This is an interesting study of a large cohort of participants, albeit with small cohorts by Covid positivity/testing which is inevitable.

The authors should comment of the country distribution in relation to the 'recruitment' process or lack of it, nor the higher proportion of women and whether these factors would bias the results, specifically as women generally have a better sense of smell than men. Can they also comment on the validation of the smells chosen and the translations into the various languages?

Overall the study makes a useful contribution in showing that simple self-testing could have significant benefit in diagnosing Covid which very respectable sensitivity and specificity by the various analyses.

The authors should comment on whether a refined version of the study will be repeated, possibly considering other countries.

Reviewer #2 (Remarks to the Author):

There appear to be a number of issues with the layout of the manuscript:

1. Unusually the referencing starts in the abstract
2. The next section is labelled "results" but contains a mixture of methods and results
3. The manuscript lacks conventional structure with the heading methods not appearing until page 20.
4. Where is the aim and the hypothesis?

The manuscript needs a major reconstruction to work as a scientific paper.

Ultimately 91.6% of participants didn't have a covid test, so to suggest they can screen for Covid-19 on the basis of the small numbers who were confirmed covid positive seems incredulous!

There is no reference to the GCCR work which has far greater numbers of covid +ve and covid -ve cases reported.

Furthermore, the time plots of odourant rating against cases for each country don't visually correlate for many of the countries depicted.

Reviewer #3 (Remarks to the Author):

The authors are exploring olfactory perception as a tool to approximate COVID-19 epidemiology and individual infections. The authors explore the idea of using such a tool as an additional diagnostic tool to tackle the ongoing pandemic. Overall the paper is very interesting and any additional tools to speed up detection of infections is important. I also appreciate that the authors address limitations up front, yet some questions remain for me in regards to usability and true accuracy. (Of note, I am not an expert of olfactory perception, hence can not comment adequately on the used methodology)

Detailed comments:

- Could the authors elaborate how applicable they expect the algorithm to be across countries/regions/cultures? Olive oil, basil, cumin are all smells/flavors that are associated with certain cultures and regions and I did not clearly understand if the authors saw a regional/cultural shift in performance?
- Similar to the above point around cultural applicability: how would such a hypothetical test need to be adapted and validated locally?
- The authors refer to symptom checkers as a tool to monitor infection (referencing a framework); is this currently used in monitoring by any surveillance system? Hence how applicable is this idea from a public health perspective?
- Please update your AgRDT estimates. See for example the most recent Cochrane systematic review for better estimates (~72% with many of the tests reaching up to 90%). In a meta analysis, some of the AgRDTs show performance above 80% and hence meet the WHO target product profile definitions for performance.
- Could the authors provide a clear concept of how they envision their diagnostic tool to work or be implemented? Even if only as a vision? A lot of different analysis are employed but it remains unclear to me how practically such a tool is envisioned to be used in a systematic way (when compared to standard IVDs like PCR/Ag)? In other words, is a comparison with such tools really appropriate?
- Overall in my opinion the section on sens/spec and diagnostic performance should be tuned down and shortened significantly. Move detail to the methods. In line with my above comment, it feels unrealistic that using such a tool as direct test in the same way as a IVD and the paper would benefit from a link to reality in terms of WHO guidance (refer to the published Target Product Profile), regulatory requirements, potential use cases and advantages (eg. cost? simplicity?)
- All these accuracy calculations are made based on self reported symptoms and test results. How has reporting bias been incorporated; I don't see this appropriately addressed. Who was the population participating in this study and how was the result confirmed? Gold standard and references are critical for diagnostic accuracy assessment and I think this is a major issue in the study that I don't feel is appropriately addressed.
- the authors note that patients without symptoms would have not been picked up by symptom checkers; yet these same patients would have not been included in the analysis as they would not have reported symptoms or test results in the checker. Why do the authors think these type of patients would have been identified based as the comparator is still a positive COVID-19 dx test. Patients without symptoms would have not presented for testing. Clarify if this has been considered in the analysis.
- Overall the ms can be significantly shortened!

Reviewer #4 (Remarks to the Author):

This research is quite interesting for me. Authors collected quite a large amount of data from Internet. They claimed that the olfactory testing would be highly effective in COVID19 detection. I have the following concerns for this study.

1. It seems that authors restricted their study to participants with symptoms only. It might be understandable that they have good sensitivity and specificity values. The question is the dataset they used was not that 'clean' that definitely bring biased data or even noise. It will greatly hurt the robustness of the method. Author should address this more though they mentioned it in discussion.

2. The popular AI COVID19 detection using cough data can achieve even much better sensitivity and specificity than the proposed method. How to address the merit of this approach compared to the AI cough data way?

3. More evidence may be needed to say olfactory testing remained highly effective in participants with COVID-19 but without symptoms, and in participants with symptoms but without COVID-19.

4. Olfactory Perceptual Fingerprint looks like a nice idea. But I doubt it can really work well. It needs 4 odorants: 'Vanilla extract', 'Garlic, freshly chopped' and 'Vinegar white'. How can you guarantee people with possible COVID symptoms find them in their kitchen? I wonder how it will be that practical

5. Authors need to employ or build a rigorous model to support their conclusion.

To all referees:

We thank you for your efforts and constructive review.

First, we would like to apologize for the time it has taken us to submit this revision. This reflected in part the collection and analysis of additional data, but also a personal issue of the project PI (NS) who managed the submission, but was away from work for three months.

Second, and to the point, **we concurred with all comments made.** This entailed modifying analyses, modifying text, and most critically, adding an additional data set of later-collected data, containing a much higher proportion of tested, and C19+ participants (consistent with the times). This independent data-set verified our model. The process of addressing the referee comments enabled us to crystalize the manuscript. We think where this tool stands out is in its ability to detect asymptomatic COVID-19 positive individuals, an ability previously attributed, as far as we know, only to various molecular methods. We hope the manuscript now satisfies the criteria for publication in Communications Medicine. Detailed reply follows, referee comments in **RED**, our reply in black:

Reviewers' comments:**Reviewer #1 (Remarks to the Author):**

This is an interesting study of a large cohort of participants, albeit with small cohorts by Covid positivity/testing which is inevitable.

We thank the Referee for the positive comment.

The authors should comment of the country distribution in relation to the 'recruitment' process or lack of it, nor the higher proportion of women and whether these factors would bias the results, specifically as women generally have a better sense of smell than men.

To address this comment, we now added in the "limitations" paragraph in the discussion: "Similarly, we observe that participants were not evenly distributed across countries, nor across sexes: more women participated than men. Although some studies have suggested that women may have a better sense of smell than men (54), we fail to see how this could influence the OFP-based analysis. This, again, is because the OFP is a relative measure, and not an absolute measure of performance per se. Therefore, these sampling biases, although unwanted, likely did not influence the reported result."

Can they also comment on the validation of the smells chosen and the translations into the various languages?

We now better reiterate in the text that the power of the olfactory fingerprint is in that no matter what odorants the participant uses, we can independently calculate the fingerprint. As to language, the terms were verified by native participating scientists for each language.

Overall the study makes a useful contribution in showing that simple self-testing could have significant

benefit in diagnosing Covid which very respectable sensitivity and specificity by the various analyses. The authors should comment on whether a refined version of the study will be repeated, possibly considering other countries.

In the revision we analyze an **added data download of 1464 new participants**. This added cohort lends significant added strength to the results. The tool is of course available world-wide, and we hope to increase languages beyond the current 15.

We thank the Referee for what was a constructive review, and we hope we have addressed it effectively.

Reviewer #2 (Remarks to the Author):

There appear to be a number of issues with the layout of the manuscript:

1. Unusually the referencing starts in the abstract
2. The next section is labelled "results" but contains a mixture of methods and results
3. The manuscript lacks conventional structure with the heading methods not appearing until page 20.

Regarding these three formatting-related comments: Our manuscript was submitted to Communications Medicine using the "transfer manuscript" function. It was initially submitted to Nature, where the style is a referenced abstract, and no introduction. The manuscript was rejected without review from Nature, but the Editors suggested we transfer the manuscript. The Nature transfer instructions are that one directly transfers, and can reformat following review. We have now reformatted accordingly. **To conclude, all of the above have been addressed in the revision.**

4. Where is the aim and the hypothesis?

This is of course always good advice to clearly state this, and we thank the referee for helping us crystalize this effort. The new introduction now ends with the following **explicit** statement: "**our hypothesis was** that the OFP may allow for accurate classification of individuals who are COVID-19 positive but without symptoms, or COVID-19 negative but with symptoms of disease. Beyond testing this hypothesis, **our aim was** to generate a convenient online tool that applies this approach." (bold added here).

The manuscript needs a major reconstruction to work as a scientific paper.

The manuscript has indeed been entirely reconstructed on the above lines.

Ultimately 91.6% of participants didn't have a covid test, so to suggest they can screen for Covid-19 on the basis of the small numbers who were confirmed covid positive seems incredulous!

We accept this point, and have now added an additional 114 COVID positive yet asymptomatic participants as part of the new cohort. This reflects a 400% increase in the ratio. Beyond this, we of course provide effect-size statistics for every analysis we report. Thus, in cases where the power of the claims is limited, this is clearly reflected in effect size. Finally, we significantly toned down the claims, using the term "screen" rather than "test" wherever appropriate.

There is no reference to the GCCR work which has far greater numbers of covid +ve and covid -ve cases reported.

This is an important criticism, and we accept in in full. We now make extensive reference to the GCCR work, and moreover, highlight the key difference between our effort and previous efforts: Clearly, a major concern in the pandemic is asymptomatic COVID-19 positive individuals. These individuals may unknowingly spread the disease. The GCCR effort did not succeed in identifying asymptomatic COVID-19 positive individuals. We verified this with our friends who are leaders in the GCCR, as well as with several co-authors here who are also part of the GCCR. The current manuscript significantly classifies these individuals using the OPF, and in this stands out. The Referee comment helped us sharpen this

point, while also leading us to add more extensive credit to the GCCR effort, now extensively detailed in the added introduction.

Furthermore, the time plots of odourant rating against cases for each country don't visually correlate for many of the countries depicted.

We note that in 4 of the 8 cases the r value is under $r = 0.3$, so this is indeed visually unimpressive. Nevertheless, these are weak correlations, but they are statistically significant. We respectfully submit that had the graph been visually impressive, **that** would be a source of concern... as an r value of 0.25 shouldn't look very impressive... It is what it is, and we tried to present it faithfully.

We thank the Referee for what was a constructive review, and we hope we have addressed it effectively.

Reviewer #3 (Remarks to the Author):

The authors are exploring olfactory perception as a tool to approximate COVID-19 epidemiology and individual infections. The authors explore the idea of using such a tool as an additional diagnostic tool to tackle the ongoing pandemic. Overall the paper is very interesting and any additional tools to speed up detection of infections is important. I also appreciate that the authors address limitations up front, yet some questions remain for me in regards to usability and true accuracy. (Of note, I am not an expert of olfactory perception, hence cannot comment adequately on the used methodology)

We thank the Referee for the positive comments.

Detailed comments:

- Could the authors elaborate how applicable they expect the algorithm to be across countries/regions/cultures? Oliver oil, basil, cumin are all smells/flavors that are associated with certain cultures and regions and I did not clearly understand if the authors saw a regional/cultural shift in performance?

This is a keen question, and it implies we did not do a good enough job in explaining our measure. We have tried to improve on that. More specifically, the power of the olfactory perceptual fingerprint (OPF) is that it relies on the relative differences from the overall mean rating. The mean is not regional, it's over all the data. Thus, if there was some huge cultural variability in these two ratings, that would only weaken our measure, it could not "help" us. However, in contrast to some popular notions, these two particular ratings (intensity and pleasantness) are in fact quite stable across large populations, and across very diverse cultures. A good recent consideration of this is in:

Arshamian, Artin, Richard C. Gerkin, Nicole Kruspe, Ewelina Wnuk, Simeon Floyd, Carolyn O'Meara, Gabriela Garrido Rodriguez, Johan N. Lundstrom, Joel Drewery Mainland, and Asifa Majid. "The perception of odor pleasantness is shared across cultures." bioRxiv (2021).

This is still a bioRxiv paper, but it is a collaboration of three field leaders (Lundstrom, Mainland, Majid), so I am confident it will soon be published, likely in a high-profile journal. Moreover, this notion was supported by several previous, albeit less comprehensive, yet published studies (1, 2). Although some odors have gained notoriety as outliers in this respect (guava, cilantro, durian), they have gained notoriety for indeed that reason: they are outliers. Otherwise we see very high agreement on these two ratings, and we have added several references on this front. To conclude, significant cultural variability on these ratings could have only hurt us, not helped us, and yet our measure works internationally (well, at least in places with internet...).

- Similar to the above point around cultural applicability: how would such a hypothetical test need to be adapted and validated locally?

As noted, it would not need to be validated locally. We compare against a universal mean rating.

- The authors refer to symptom checkers as a tool to monitor infection (referencing a framework); is this currently used in monitoring by any surveillance system? Hence how applicable is this idea from a public health perspective?

Yes, a symptom-checker system is being used by the national health system in Israel (Rossman, Hagai, Ayya Keshet, Smadar Shilo, Amir Gavrieli, Tal Bauman, Ori Cohen, Esti Shelly et al. "A framework for identifying regional outbreak and spread of COVID-19 from one-minute population-wide surveys." *Nature Medicine* 26, no. 5 (2020): 634-638). We do not know off-hand of additional places, but would not be surprised.

- Please update your AgRDT estimates. See for example the most recent Cochrane systematic review for better estimates (~72% with many of the tests reaching up to 90%). In a meta analysis, some of the AgRDTs show performance above 80% and hence meet the WHO target product profile definitions for performance.

We thank the referee for this comment, and have updated accordingly. We also removed any comparison to AgRDT from the abstract. As a rule, in science (and in life:-), one should indeed refrain from highlighting the weakness of others as a path to strengthening oneself... We may have erred this way in the previous version, and have tried to amend here. We thank the referee for keeping us on track.

- Could the authors provide a clear concept of how they envision their diagnostic tool to work or be implemented? Even if only as a vision? A lot of different analysis are employed but it remains unclear to me how practically such a tool is envisioned to be used in a systematic way (when compared to standard IVDs like PCR/Ag)? In other words, is a comparison with such tools really appropriate?

We sincerely thank the referee for this question that served to focus our writing and work. With this in mind, we have now modified our tool such that it tells the participant our probability estimation of his/her having COVID-19. This is now reflected in added Figure 6. I (NS) would like to share with the Referee an anecdote: A close friend of mine told me he wasn't feeling too good. I asked him if he felt a change in smell or taste. He said "NO". I told him, log onto SmellTracker, and test yourself. He did, and our new system told him we estimate he has an 82% chance of being COVID-19 positive. He therefore went to be tested, and was indeed positive. Thus, a person who didn't notice that he lost/changed smell, was still detected by our system. This is merely an anecdote, but I report this here to clarify how we envision this contribution which has now been much sharpened.

- Overall in my opinion the section on sens/spec and diagnostic performance should be tuned down and shortened significantly. Move detail to the methods. In line with my above comment, it feels unrealistic that using such a tool as direct test in the same way as a IVD and the paper would benefit from a link to reality in terms of WHO guidance (refer to the published Target Product Profile), regulatory requirements, potential use cases and advantages (eg. cost? simplicity?). All these accuracy calculations are made based on self reported symptoms and test results. How has reporting bias been incorporated; I don't see this appropriately addressed. Who was the population participating in this study and how was the result confirmed? Gold standard and references are critical for diagnostic accuracy assessment and I think this is a major issue in the study that I don't feel is appropriately addressed.

We fully acknowledge and embrace the essence of this comment, namely that we do not yet here provide a clinical tool per se. Indeed, we are mostly basic olfaction scientists, not clinicians or epidemiologists, and this shows in this respect. We now clearly acknowledge this explicitly in the manuscript. First, we edited the entire manuscript to use the term "screen" rather than "test", from

abstract and on. Second, in the discussion we now clearly state: " This study has several limitations. First, we should clearly acknowledge that this study is firstly a basic science effort, and not a clinical effort in the classic sense. For example, we do not address WHO guidelines or Target Product Profile (53), nor do we address any Gold-Standards of diagnostics. What we do here is to find that the OPF allows classification even in asymptomatic individuals, and we provide an online tool that may assist in this. Much more work is needed for this effort to satisfy clinical standards". Beyond all this, as noted repeatedly in this document, we have significantly toned down the manuscript, mostly now referring to "screening" rather than "diagnosis".

- the authors note that patients without symptoms would have not been picked up by symptom checkers; yet these same patients would have not been included in the analysis as they would not have reported symptoms or test results in the checker. Why do the authors think these type of patients would have been identified based as the comparator is still a positive COVID-19 dx test. Patients without symptoms would have not presented for testing. Clarify if this has been considered in the analysis.

Respectfully, we are not sure we understand this question. Patients without symptoms are indeed included in symptom-checker performance analyses, and here too, we had a symptom checker, and thousands of participants without symptoms completed it, including 114 C19+ in the added data.

- Overall the ms can be significantly shortened!

The manuscript is now well within journal word-count.

We thank the Referee for what was a constructive review, and we hope we have addressed it effectively.

Reviewer #4 (Remarks to the Author):

This research is quite interesting for me. Authors collected quite a large amount of data from Internet. They claimed that the olfactory testing would be highly effective in COVID19 detection.

We thank the Referee for the positive comments.

I have the following concerns for this study.

1. It seems that authors restricted their study to participants with symptoms only. It might be understandable that they have good sensitivity and specificity values. The question is the dataset they used was not that 'clean' that definitely bring biased data or even noise. It will greatly hurt the robustness of the method. Author should address this more though they mentioned it in discussion.

This comment implies that we failed to write clearly, as this is incorrect. More specifically: 55.3% of C19- participants, and 55% of C19-UD participants, had absolutely no symptoms. Moreover, the added cohort of 1464 participants is entirely asymptomatic, including 114 asymptomatic C19+ participants. We are sorry that we somehow failed to convey this effectively in the previous version, and thank the referee for leading us to highlight this.

2. The popular AI COVID19 detection using cough data can achieve even much better sensitivity and specificity than the proposed method. How to address the merit of this approach compared to the AI cough data way?

First, we thank the referee for pointing at this, and we have now referred to this in the manuscript. Second, where we think our effort stands out is in the test of completely asymptomatic individuals. As far as we know, AI COVID19 detection using cough data cannot classify C19+ individuals who have no symptoms (e.g., cough or hoarse voice). That said, in no way do we think that one method is “the way to go”, and our method may provide but an added tool in the arsenal.

3. More evidence may be needed to say olfactory testing remained highly effective in participants with COVID-19 but without symptoms, and in participants with symptoms but without COVID-19.

As noted above, the revised manuscript contains exactly such an addition of data. We thank the referee for leading us in this direction.

4. Olfactory Perceptual Fingerprint looks like a nice idea. But I doubt it can really work well. It needs 4 odorants: 'Vanilla extract', 'Garlic, freshly chopped' and 'Vinegar white'. How can you guarantee people with possible COVID symptoms find them in their kitchen? I wonder how it will be that practical

Finger prints can be odorant-based or descriptor-based. The odorant-based olfactory fingerprint does not need those specific 4 odorants, it can be calculated with any set of odorants, as long as we have enough people in the dataset that used that specific set of 4. Thus, if enough people will use SmellTracker, we can use fingerprints regardless of odorants. Nevertheless, partly with this comment in mind, we switched to an alternative version of the fingerprint, the descriptor-based version. This version

works across odorants. As the data imply, this works effectively, including a test on novel data that was not used for training.

5. Authors need to employ or build a rigorous model to support their conclusion.

As noted, the model now tested on independent data.

We thank the Referee for what was a constructive review, and we hope we have addressed it effectively.

1. Haddad R, Medhanie A, Roth Y, Harel D, & Sobel N (2010) Predicting odor pleasantness with an electronic nose. *PLoS computational biology* 6(4):e1000740.
2. Khan RM, *et al.* (2007) Predicting odor pleasantness from odorant structure: pleasantness as a reflection of the physical world. *Journal of Neuroscience* 27(37):10015-10023.

Reviewers' comments:

Reviewer #1 (Remarks to the Author):

The authors have, in my view, adequately addressed the reviewers' comments, in as much as they are able to do so. The only minor remark relates to the use of colloquial abbreviations ie don't etc which should be written in full ie do not.

Reviewer #2 (Remarks to the Author):

The abstract lacks details of how many respondents were positive. The last line of the abstract is missing a word but more importantly lacks any sense of purpose - how do the authors see this working in the real world?

The introduction contained in depth details of discussion around AI that is more appropriate for the discussion section. the introduction should set the scene and explain why you have undertaken the study. Comparison with other work is part of the discussion. I would counsel the authors to refer to the STROBE guidelines for reporting a study.

"a large consortium of laboratories" is not an appropriate description for the GCCR given there is a mixture of clinicians and non-clinicians in the group.

The presence of olfactory dysfunction in isolation has been reported already:
<https://doi.org/10.1111/coa.13683>

The methods need a lot of work in terms of its layout - again refer to the STROBE guidelines for this. Methods should set out what was intended at the beginning; variation from this such as the "india" issue should be described in the results. Describe limitations of the methods under the limitations section in the discussion.

Methods are listed in the results and the results venture into discussion

The manuscript is is too effusive and needs to present the information in a more concise manner, keeping statistical results in tables with the text summarising the key data. The use of the STROBE guidelines will help to provide this structure which is currently lacking throughout.

Reviewer #4 (Remarks to the Author):

It seems that the updated version addressed almost all my concerns. I am happy to see this method proposed in this paper can work together with COVID AI diagnosis to detect COVID.

I vote for 'acceptance' for this manuscript.

Reviewer #5 (Remarks to the Author):

The authors prepared a web-based tool for screening of olfactory loss. I have read their replies to the previous reviewers' comments and think they did a good job in most cases. I would still suggest a few minor but important text edits:

1. Loss of smell in vaccinated individuals is far less prominent than in unvaccinated. This means that real-life applicability of the tool regarding implementation in any public health setting by now is unlikely. It still remains a nice study, but this important limitation must be presented and the frequency of olfactory alterations in vaccinated vs. unvaccinated individuals, and across age groups (where possible).

2. I am weary about comparing antigen test sensitivity and specificity across countries. Particularly in resource-limited settings, antigen test reliability can vary dramatically (see, e.g., Haage et al. *J Clin Virol* 2021).

3. Previous rev. 3 has a good point in comparing results between individual countries. The authors provide a good reply that should be incorporated into the manuscript.

Dear Editor and Referees

We thank you for the re-review, we concur with all comments, and have edited the manuscript accordingly

Details follow:

Reviewer #1 (Remarks to the Author):

The authors have, in my view, adequately addressed the reviewers' comments, in as much as they are able to do so. The only minor remark relates to the use of colloquial abbreviations ie don't etc which should be written in full ie do not.

We thank the Referee for this comment, and have corrected throughout the manuscript. We thank the referee for their positive conclusion.

Reviewer #2 (Remarks to the Author):

The abstract lacks details of how many respondents were positive.

This number, 462, has now been added to the abstract

The last line of the abstract is missing a word but more importantly lacks any sense of purpose - how do the authors see this working in the real world?

We acknowledge that this is an important question, but it is not one we can address in the abstract, given abstract space limitations. In view of this comment echoed in the previous review cycle, we note at the end of the results section:

"With these results in hand, we modified the feedback component of the online tool, and this now allows for a more effective implementation. More specifically, a user can now participate using any 5 of the 71 available odorants, and after completing the perceptual estimates, we inform the user to what extent (in percentages) he/she resembles a C19+ or a C19- individual (Figure 6). A person who has high resemblance to C19+ individuals, should likely minimize contact with others, and seek a formal test."

The introduction contained in depth details of discussion around AI that is more appropriate for the discussion section. the introduction should set the scene and explain why you have undertaken the study. Comparison with other work is part of the discussion. I would counsel the authors to refer to the STROBE guidelines for reporting a study.

We have streamlined the information across introduction and discussion

"a large consortium of laboratories" is not an appropriate description for the GCCR given there is a mixture of clinicians and non-clinicians in the group.

We have changed the wording to: "a large consortium of clinicians and basic scientists known as GCCR"

The presence of olfactory dysfunction in isolation has been reported already: <https://doi.org/10.1111/coa.13683>

The above study indeed highlights where ours stand out: Whereas the above was a questionnaire asking people to report their symptoms, our study actually had participants smell real odorants and rate them. This allowed us to identify participants who were unaware of their loss.

The methods need a lot of work in terms of its layout - again refer to the STROBE guidelines for this. Methods should set out what was intended at the beginning; variation from this such as the "india" issue should be described in the results. Describe limitations of the methods under the limitations section in the discussion. Methods are listed in the results and the results venture into discussion. The manuscript is too effusive and needs to present the information in a more concise manner, keeping statistical results in tables with the text summarising the key data. The use of the STROBE guidelines will help to provide this structure which is currently lacking throughout.

We have moved various statements to maintain correct manuscript structure.

We thank the referee for their constructive advice.

Reviewer #4 (Remarks to the Author):

It seems that the updated version addressed almost all my concerns. I am happy to see this method proposed in this paper can work together with COVID AI diagnosis to detect COVID.

I vote for 'acceptance' for this manuscript.

We thank the referee for their positive conclusion.

Reviewer #5 (Remarks to the Author):

The authors prepared a web-based tool for screening of olfactory loss. I have read their replies to the previous reviewers' comments and think they did a good job in most cases. I would still suggest a few minor but important text edits:

1. Loss of smell in vaccinated individuals is far less prominent than in unvaccinated. This means that real-life applicability of the tool regarding implementation in any public health setting by now is unlikely. It still remains a nice study, but this important limitation must be presented and the frequency of olfactory alterations in vaccinated vs. unvaccinated individuals, and across age groups (where possible).

This is a keen observation, that was of course irrelevant when we first submitted, as that was before vaccinations were available. There is yet a peer-reviewed paper that we are aware of

that discusses the reduced olfactory loss in the vaccinated, but we have now added this to the discussion:

"Beyond all of the above, the COVID-19 pandemic is rapidly evolving, in fact far faster than the process of publishing a manuscript. For example, when we collected the first portion of the data reported here, COVID-19 tests were barely available, yet when we collected the second portion of the data, tests were very common, which made for the higher proportion of COVID-19+ participants. In turn, even at the second collection stage, vaccines were not yet widely available, yet now they are. Anecdotal evidence suggests reduced olfactory loss in vaccinated individuals that nevertheless contract the disease. If these anecdotes materialize, this will render our tool not useful for the vaccinated. This, however, awaits formal results on olfaction in sick yet vaccinated individuals."

2. I am weary about comparing antigen test sensitivity and specificity across countries. Particularly in resource-limited settings, antigen test reliability can vary dramatically (see, e.g., Haage et al. J Clin Virol 2021).

We concur, and have added this reference to the manuscript as follows:

"Antigen results vary widely³⁶, and their results are environmentally dependent³⁷."

3. Previous rev. 3 has a good point in comparing results between individual countries. The authors provide a good reply that should be incorporated into the manuscript.

With this comment in mind, we have taken the previous Referee 3 comment, and our answer, and added them combined as a new paragraph in the discussion:

"To readers not versed in human olfaction, it may be unclear how our algorithm retains performance across countries, regions, and cultures. The power of the olfactory perceptual fingerprint is that it relies on the relative differences from the overall mean rating. The mean is not regional, it's over all the data. Thus, if there was some huge cultural variability in these two ratings, that would weaken our measure. However, in contrast to some popular notions, these two particular ratings (intensity and pleasantness) are in fact quite stable across large populations, and across very diverse cultures^{15,16,20,29,30}. Although some odors have gained notoriety as outliers in this respect (guava, cilantro, durian), they have gained notoriety for exactly that reason: they are outliers. Otherwise we see very high agreement on these two ratings^{15,16,20,29,30}."

We thank the referee for their positive conclusion.